# Cultivation potential projections of breadfruit (*Artocarpus altilis*) under climate change scenarios using an empirically validated suitability model calibrated in Hawai'i

**Kalisi Mausio**[1], **Tomoaki Miura**[2], **Noa K. Lincoln**[1]*

**1** Tropical Plant and Soil Sciences, University of Hawai'i at Manoa, Honolulu, HI, Unites States of America, **2** Natural Resources and Environmental Management, University of Hawai'i at Manoa, Honolulu, HI, Unites States of America

☯ These authors contributed equally to this work.
* nlincoln@hawaii.edu

**Data Availability Statement:** All model codes and data layer files are available at GitHub for public access: https://github.com/nlincoln2017/Breadfruit-Suitability-Model

## Abstract

Humanity faces significant challenges to agriculture and human nutrition, and changes in climate are predicted to make such challenges greater in the future. Neglected and underutilized crops may play a role in mitigating and addressing such challenges. Breadfruit is a long-lived tree crop that is a nutritious, carbohydrate-rich staple, which is a priority crop in this regard. A fuzzy-set modeling approach was applied, refined, and validated for breadfruit to determine its current and future potential productivity. Hawai'i was used as a model system, with over 1,200 naturalized trees utilized to calibrate a habitat suitability model and 56 producer sites used to validate the model. The parameters were then applied globally on 17 global climate models at the RCP 4.5 and RCP 8.5 global climate projections for 2070. Overall, breadfruit suitability increases in area and in quality, with larger increases occurring in the RCP 8.5 projection. Current producing regions largely remain unchanged in both projections, indicating relative stability of production potential in current growing regions. Breadfruit, and other tropical indigenous food crops present strong opportunities for cultivation and food security risk management strategies moving forward.

## Introduction

Humanity faces multiple challenges for the future of food production. By mid-21st century, the world population is expected to reach nine billion people with associated pressure on resources [1], increasing demands on food and nutrition while rates of hunger and malnutrition are on the rise [2]. Changes in global weather are expected to negatively impact food yields, especially the major commodity crops that provide much of the global food supply [3,4], and nutrient quality and density of crops [5,6]. Global hunger and malnourishment, largely correlated to poverty and insufficient access to enough nutritious food is increasing [2,7]. Some 2 billion people are suffering from micronutrient malnutrition [8,9]. At the other

**Funding:** NKL have received McIntire-Stennis (8038-MS), HATCH (8035-H) and Western SARE (SW17-050) funding that has supported this project. The funders had no role in study design, data collection and analysis, decision to publish, or preparation of the manuscript.

**Competing interests:** The authors have declared that no competing interests exist.

end of the spectrum, obesity—which is often attributed to global commodities that lack essential micro and macro-nutrients—is also on the rise [2].

Increased pressure on food markets is expected to exacerbate existing issues of food security, nutrition, social equity, and economies [1,3]. In order to achieve the goals of eliminating hunger, improving food security and nutrition and promoting sustainable agriculture, many advocates for transforming and developing food systems that respond to or anticipate climate change [10]. With existing problems in our current food supply and compounding factors on the horizon, developing food production systems that emphasize resilience and nutrition is an essential strategy for mitigating these issues.

Of the approximately 250–300,000 flowering plant species, at least 50,000 are edible [11,12]. Of these, about 3,000 species are regularly exploited for food [13]. Yet only three crops provide 60% of the world's calories [14] and 103 crops provide some 90% of the world's food production [15]. This gap between the 3,000 species regularly utilized for food and the 103 dominant crops of the planet make up a vast, relatively untapped resource of crops commonly referred to as neglected and underutilized species (NUS). Such crops are often ignored by researchers, policy makers and breeders [13,16]. NUS offer opportunities for adaptation because they have a wide range of genetic diversity that can enhance resilience to stressors related to climate change [17].

Although emphasis on NUS will undoubtedly help to mitigate climate impacts to current food systems and positively impact food and nutrition security, multiple barriers exist to their widespread adoption. These barriers include loss of genetic diversity and traditional knowledge, undervaluation through lack of knowledge and research, poor competitiveness, and lack of infrastructure, policy support and investments [16]. Therefore, more attention must be given to NUS on all levels—from cultivation to gastronomic uses—if they are to be useful in creating sustainable food systems for the future.

Breadfruit *(Artocarpus altilis)* is an underutilized tree crop that grows in the tropics and sub-tropics, originally cultivated from the wild species ancestor bread nut *(Artocarpus camansi)* in Papua New Guinea [18,19]. The crop is listed as one of the thirty-five priority crops in the International Treaty on Plant Genetic Resources for its potential to target food security and interdependence [20]. The trees are a high yielding starchy staple crop, producing up to 10 t/ha of fruit that is rich in carbohydrates, essential amino acids, fiber, vitamins, minerals including micronutrients such as iron and potassium [21–24]. Although exact fruit yield estimations are variable (see [25] for a summary of yield), it is evident that breadfruit has the potential to target issues surrounding food and nutrition security if specific cultivar and growing conditions are optimized.

Recent advances in propagation methods, scientific research and promotion of breadfruit have made breadfruit trees widespread throughout the tropics however, they are still underutilized due to the range of factors identified for NUS in general [18, 25]. As a perennial tree, breadfruit has significant potential to contribute to building climate resilient food systems [26,27] in addition to their ability to sequester carbon, provide shade, stabilize the soil, benefit watersheds, and provide a multitude of invaluable environmental benefits [28,29].

GIS-based land-use suitability analysis attempts to identify the most appropriate spatial pattern for land uses according to specific requirements, preferences, or predictors of some activity [30], including suitability of land for agricultural activities [31]. A considerable amount of research has been conducted on assessing the relationship between crop response/productivity and climate using simulation models [32,33]. Recently this approach has also been utilized as a method with which to assess crop response to climate change [34–37]. However, most of these models are fine-tuned towards predicting crop yields for global commodity crops (e.g. corn, wheat, soybean, rice), for which widespread cultivation offer ample opportunities for model

calibration and validation. The methods are quite adaptable to different crop types as long as basic information about the crop's environmental range is known.

GIS-based suitability analysis can address multiple-criteria decision-making problems and can incorporate fuzzy logic techniques [31]. Fuzzy logic [38] involves the concept of membership function in which a given element is numerically represented by the degree to which it belongs to a set. In this way, a measurement within a criterion can have degrees of membership between unsuitable (0) and perfectly suitable (1). Because geographical phenomena tend to exhibit continuous spatial variability, Burrough and McDonnell [39] suggest that fuzzy membership more accurately captures boundaries between land suitability classes than binary or categorical approaches. Such application in crop modeling has been demonstrated more recently [37].

Different approaches to modeling have been established, demonstrating methodologies that can be applied to a wide range of crops. For instance, AquaCrop is a water-driven crop growth model that utilizes linear proportionality to transpiration, with crop/cultivar specific scalar parameters [40,41]. Other models, such as CropSyst, use several dozen crop input parameters to simulate the production rate [42,43]. Driven by daily weather inputs, such models can be highly accurate when properly calibrated and have recently been applied to NUS [44–48]. However, such calibration typically requires experimental investigation over the lifetime of a crop, which in the case of long-lived tree crops, and in particular neglected and underutilized tree crops, can be prohibitive. In the case of trees, landscape-level approaches utilizing natural experiments may be better suited to understanding patterns of potential productivity.

Traditionally, breadfruit in Hawai'i was cultivated as a major staple [49–51] in a range of cropping systems, from massive arboriculture developments, to mixed agroforestry, to individual and backyard trees [52]. Following European colonization, a dramatic shift away from traditional crops occurred [53], although many pockets of traditional agriculture and associated practices remained [54] and remnant trees and production systems persisted [50, Lincoln *in press*]. Over the past 20 years, significant efforts have occurred to revitalize breadfruit in Hawai'i. Such efforts have included large-scale tree giveaways [55], restoration of traditional agricultural systems [54], a growing local food producing sector [56], and consumer education such as outreach, chef campaigns and festivals [57]. Such initiatives have resulted in significant increases in production at multiple levels including backyard trees, small diversified farm plantings, and larger mono-cropped orchards [55,56].

Hawai'i is an excellent location to tune habitat distribution models, including for agricultural species and activities. Hawaii provides a "model system" for ecological investigations [58] due to its consistent geology and wide, well-defined variation in climate and substrate age that allow for a degree of precision that cannot be duplicated elsewhere. Environmental gradients in mean annual temperature (from <10–24 C), annual precipitation (from <200->10,000 mm), and substrate age (from hot rock to >4,000 kyr) are among the clearest, broadest, and most orthogonal on Earth [59]. The matrix of environmental gradients creates among the densest concentration of ecosystems on the planet [60], and correspondingly dense variation in agricultural habitats and opportunities.

The purpose of this paper is to develop an empirically validated, fuzzy-set model for breadfruit production that incorporates both climate and soil data, and to explore the global potential for breadfruit cultivation in current and future climate scenarios. This builds upon prior work that conducted a two-tiered suitability model for breadfruit based on rainfall and temperature [61]. The developed model is utilized to further assess potential changes in breadfruit production over time with anticipated climate scenarios to understand global and regional changes in productive potential.

## Methods

### Modeling approach

The crop model methodology in this study utilizes a basic mechanistic approach in which 1) environmental criteria deemed important to the crop's success are selected, 2) suitable environmental ranges for the crop are determined for each of the selected criteria, 3) fuzzy sets are constructed for each criteria based on the environmental ranges, representing the approximate niche of the crop, 4) a crop suitability score is calculated based on how closely the current or future environmental conditions match the constructed fuzzy sets, and 5) model is validated and refined in an iterative process as required. This approach allows flexibility within the model and the ability to add new evaluation criteria.

The initial environmental ranges for breadfruit were obtained from the EcoCrop database [62] which contained an optimum range and an absolute range for each parameter. Environmental criteria with values within the optimal range represent perfect suitability while values outside the absolute range are considered unsuitable. Fuzzy sets were constructed using the methodology adapted from Ramirez-Villegas et al. [37] which utilized a linear algorithm to systematically construct the fuzzy set used to derive scores (0–100) between suitable and unsuitable:

$$SUIT_i = if[P_i < ABS_{min}, 0$$
$$ABS_{min} \leq P_i < OPT_{min}, (P_i - ABS_{min})/(OPT_{min} - ABS_{min}) * 100$$
$$OPT_{min} \leq P_i < OPT_{max}, 100$$
$$OPT_{max} \leq P_i < ABS_{max}, (1 - (P_i - OPT_{max})/(ABS_{max} - OPT_{max})) * 100$$
$$P_i > ABS_{max}, 0)]$$

where SUIT is the suitability score, and P is the measured environmental criterion at the site (i.e., each pixel), and $ABS_{min}$, $ABS_{max}$, $OPT_{min}$, and $OPT_{max}$ are the absolute and optimal environmental ranges for the criterion of a particular crop.

Five environmental criteria essential to breadfruit growth were selected to evaluate breadfruit crop suitability based on the EcoCrop parameters, availability of data, and the relationship to tree distribution. These criteria were rainfall, average temperature, solar radiation, soil drainage, and soil pH. The overall suitability score for each crop was calculated on a per pixel basis using the minimum value of the sets the cell location belongs to:

$$SUIT_{overall} = min(C_{SUIT\ 1}, C_{SUIT\ 2}, \ldots, C_{SUIT\ n})$$

where $n$ is the number of criteria used in the evaluation. This conservative approach is based on the law of the minimum [63], in which crop yield is proportional to the most limiting nutrient; the same idea can be applied to environmental conditions and has been utilized in similar GIS crop-suitability analysis [64].

Spatial data layers of the environmental parameters were obtained from the Hawaii Rainfall Atlas [65], the Hawaii Evapotranspiration Atlas [66] and the SSURGO database [67]. All calculations were performed in R Studio (RStudio, Inc., Boston, MA) [68] at a mapping unit of 50m by 50m. The R Studio "raster" and "rgdal" packages were used to generate the equations within and between the spatial layers. The scripts, layers, and imagery are available online at https://github.com/nlincoln2017/breadfruit-suitability-model.

## Model refinement and validation

Based on previous modeling of breadfruit habitat [62, 57], the EcoCrop environmental parameters were known to not be accurate. We addressed this by overlaying the habitat suitability maps generated from the EcoCrop parameters with a map of 1,200 naturalized breadfruit trees from systematic surveys of breadfruit on four islands (Kaua'i, Molokai, O'ahu, and Hawai'i) (e.g., [50, 55]); and convening a panel of experts to discuss and refine the optimal and absolute levels for each of the environmental parameters. The model was regenerated with adjusted absolute and optimal environmental parameters for two iterations at which point the panel of experts agreed upon the ranges and the resulting model closely aligned with the mapped tree distributions.

To validate the new model, yield and productivity data from 56 producer sites were used (see [69] for details on producer sites and methods of quantifying productivity). These producer sites were categorized as follows: 11 both irrigated and amended soils, 15 irrigated only, 4 amended soils only, and 27 neither irrigated nor amended soils. Model validation was conducted by comparing the modeled habitat suitability to the observed breadfruit productivity, both utilizing a scale of 0–100.

The model efficacy was evaluated using Root Mean Square Error-observations standard deviation ratio (RSR), Nash-Sutcliff Model Efficiency coefficient (ME) [69], and Willmott's Index of Agreement (IA) [70] using the following equations:

$$RSR = \sqrt[2]{1 - \frac{\sum_{i=1}^{n}(M_i - S_i)^2}{\sum_{i=1}^{n}(M_i - \bar{M})^2}}$$

$$ME = 1 - \frac{\sum_{i=1}^{n}(M_i - S_i)^2}{\sum_{i=1}^{n}(M_i - \bar{M})^2}$$

$$IA = 1 - \frac{\sum_{i=1}^{n}(S_i - M_i)^2}{\sum_{i=1}^{n}(|S_i - \bar{M}| + |M_i - \bar{M}|)^2}$$

Where $S_i$ and $M_i$ are the simulated and measured values of yield and $\bar{M}$ is the average of $M_i$ values of $n$ measured values. The RSR is an indicator of the distance between the observed and simulated values; the closer the value is to zero the better the model simulation. The ME measures the departure of the model compared to the observed variance, where ME = 1 indicates a perfect model fit and ME = 0 means that the observed mean value is as good a predictor as the model. The IA measures the ratio of the mean square error to the total potential error, with IA = 1 indicates a perfect fit and IA = 0 represents the worst possible model. Values were calculated for all sites against corresponding model output of combined environmental parameters. For instance, modeled outputs compared to irrigated sites did not include the rainfall parameter since this is not a limiting parameter for the irrigated sites. The same concept was applied for the soil pH parameter and sites that used soil amendments. At sites where trees were neither irrigated nor amended, the model output that used all environmental parameters was applied.

Following the development, refinement, and validation of the model in Hawai'i, the derived ranges of optimal and absolute environmental conditions were used to run the model at a global level. The environmental layers of mean annual temperature, rainfall and solar radiation were acquired from WorldClim [71] and the global soil data of pH and drainage class from the

WISE database [72,73]. A comparison of the global model to a previous model developed by Lucas and Ragone [62] was conducted by spatially overlaying the two datasets and extracting spatially corresponding suitability scores. A mosaic plot was generated to anecdotally examine the similarities and differences in predicted extent and quality of breadfruit habitat between the two models.

### Future projections

Global average annual temperature and rainfall projection data from the WorldClim database were applied to our model to create future suitability scores. These datasets represent the 2060–2080 average using global climate models (GCM) from the CMIP5 of the IPPC Fifth Assessment that were downscaled to a 30 arc second (~1km at the equator) spatial resolution using current WordClim 1.4 as baseline current (1950–2000) data [74]. All 17 GCMs which had projections of the environmental parameters for both Representative Concentration Pathway (RCP) 8.5 and 4.5 —representing extreme and intermediate Greenhouse Gas scenarios–were applied. Each set of projections were used to generate breadfruit suitability using our validated breadfruit model. The 17 model outputs from the GCM scenarios were averaged to represent 2070 breadfruit suitability for the two RCPs. Finally, each RCP scenario was compared to the current global suitability output to determine how much suitability would increase or decrease in the next 50 years by spatially overlaying the two datasets and extracting corresponding suitability scores.

## Results and discussion

### Hawai'i model and validation

The initial breadfruit model that utilized parameters defined by EcoCrop proved to be highly restrictive compared to the distribution of naturalized trees surveyed in Hawai'i especially in terms of rainfall and soil drainage (Table 1). For instance, the absolute maximum rainfall as defined by EcoCrop was 3,500 mm/yr, but tree mapping in Hawai'i demonstrated naturalized trees in substantially wetter areas and experts provided numerous contrary examples from across the Pacific. Similarly, there were ample trees documented in Hawai'i growing on higher and lower drainage class than reported by EcoCrop. Less significant changes were recommended for temperature and pH, and no changes to solar radiation were suggested. Final parameters utilized were in all cases less restrictive than provided by EcoCrop (Table 1).

Following refinement of the model parameters, a fuzzy set model was produced for Hawai'i (Fig 1). The effects of Hawai'i's substantial gradients in temperature and rainfall are clearly visible, with greater potential on the wetter windward (northeastern) sides of the islands and near

**Table 1. Environmental parameters initially obtained from the EcoCrop database and the refined parameters applied to the habitat suitability model.**

|  |  | Abs Min | Abs Max | Opt Min | Opt Max |
|---|---|---|---|---|---|
| **Final Model Parameters** | **Temp** (°C) | 17 | 40 | 21 | 33 |
|  | **Rain** (mm/yr) | 750 | 8000 | 1500 | 4000 |
|  | **Solar Rad.** (W/m$^2$) | 20 | 295 | 50 | 197 |
|  | **pH** | 4 | 8.7 | 5 | 6.5 |
|  | **Drainage Class** | 2 | 7 | 4 | 6 |
| **EcoCrop Parameters** | **Temp** (°C) | 16 | 40 | 21 | 33 |
|  | **Rain** (mm/yr) | 1000 | 3500 | 1500 | 3000 |
|  | **Solar Rad.** (W/m$^2$) | 20 | 295 | 50 | 197 |
|  | **pH** | 4.3 | 8.7 | 5.5 | 6.5 |
|  | **Drainage Class** | 4 | 6 | 4 | 6 |

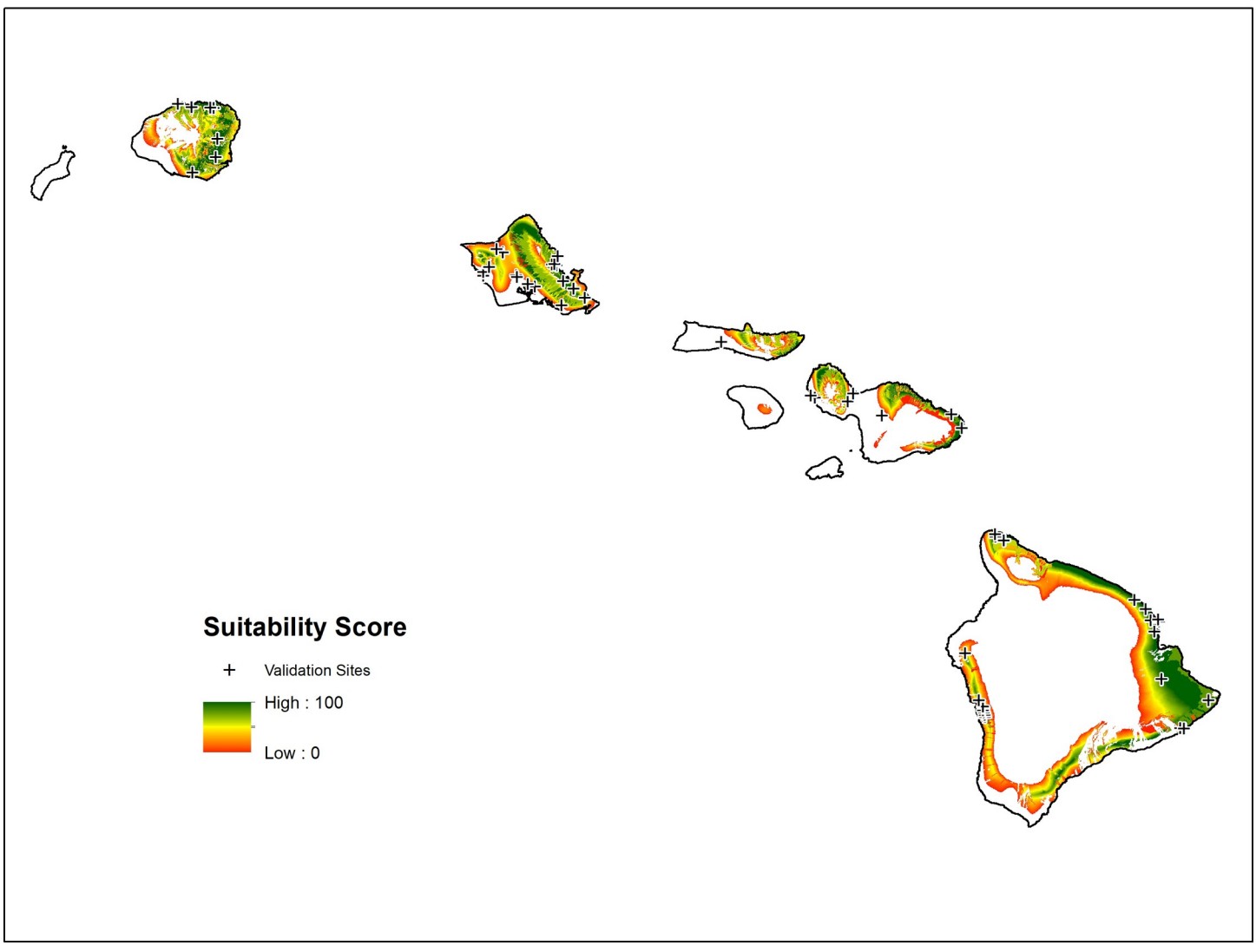

**Fig 1. Suitability for productive capacity of breadfruit in Hawai'i, with scores ranging from 0 (white, cannot cultivate) to green (100, ideal cultivation) based on 5 climate and soil parameters.** Validation sites are marked by the black crosshairs.

the warmer coasts. The textured breaks in the model represent the soil parameters driven by different aged lava flows on the younger (southern) islands and valley topography on the older (northern) islands. Leveraging the diversity across the archipelago, validation was conducted using sites located on five islands (Fig 1).

The sites used for validation differed substantially in both their actual productivity and their modeled suitability. A linear regression between measured productivity of "natural" sites (no irrigation and no soil amendments) and corresponding simulated suitability demonstrated a strong, significant relationship ($r^2$ 0.91, p<0.001). Using all sites and their corresponding simulated suitability the relationship weakens but remains strong ($r^2$ 0.84, p<0.001). Model validation statistics indicated that overall our model performs very well with moderately high model efficiency (ME) and very high index of agreement (IA) (Table 2). It is suggested that models perform satisfactorily if ME > 0.5 and the RSR is below 0.7 [75], with our model performing substantially stronger.

**Table 2. Summary of validation site statistics in terms of measured and simulated productivity and their coefficient of variation, and model accuracy assessment for only sites without irrigation or fertilization and for all sites.**

| Irrigated | Fertilized | n | Meas. Prod. | Meas. CV | Sim. Prod. | Sim. CV | Model Used |
|---|---|---|---|---|---|---|---|
| N | N | 27 | 65.7 | 28.9 | 63.2 | 38.4 | All model parameters |
| N | Y | 15 | 78.7 | 34.4 | 81.9 | 50.2 | Removed rainfall consideration |
| Y | N | 4 | 80.0 | 24.8 | 59.5 | 29.1 | Removed pH consideration |
| Y | Y | 11 | 92.7 | 25.3 | 99.5 | 10.7 | Removed rainfall and pH consideration |
| Y/N | Y/N | 57 | 75.4 | 8.5 | 74.9 | 1.7 | Relevant model for each site |
| **Irrigated** | **Fertilized** | **n** | **RMSE** | **St Dev** | **ME** | **IA** | **Model Used** |
| N | N | 27 | 12.2 | 22.6 | 0.70 | 0.95 | All model parameters |
| Y/N | Y/N | 57 | 12.5 | 21.7 | 0.67 | 0.94 | Relevant model for each site |

## Global model and comparison to previous model

Following the development, refinement, and validation of the model using Hawai'i as a model system, the parameters were applied to global data layers to make global predictions of production potential (Fig 2).

We compared our empirically validated model against the only previous global breadfruit suitability model published (Fig 3), which applied only rainfall and temperature to define two classes of breadfruit habitat: "suitable" and "best" [62]. Overall our model is more inclusive, with the model by Lucas and Ragone [62] suggesting ~26,900,000 km$^2$ of suitable habitat (~14,800,000 km$^2$ of "best" and ~12,100,000 of "suitable"), and our model suggesting ~35,100,000 km$^2$ of suitable habitat–an ~30% increase in total cultivable area. This is likely due to our more inclusive thresholds for rainfall and temperature as defined in the model fitting process. Since our model applies the law of minimums, the inclusion of additional criteria should have further restricted the extent of breadfruit habitat. Of the lands excluded by Lucas and Ragone [62] and included by our model, ~70% of those lands score 50 or below on the fuzzy set; however, ~10% do show exceptional suitability, scoring 90 or above. Of the lands included by Lucas and Ragone [62], our model excludes ~1,300,000 km$^2$ of "suitable" lands and 700,000 km$^2$ of "best" lands. A brief examination of excluded pixels indicates that these exclusions primarily resulted from the consideration of soil parameters in our model. Overall, however, the two models show fairly good alignment. Of the "best" lands identified by Lucas and Ragone [62], ~90% of the points demonstrate suitability values of 90 or above by our model. The average simulated suitability by our model of the "best" lands in Lucas in Ragone [62] is 89, for "suitable" lands 61, and for "unsuitable" lands 0.6.

## Future projections

Future breadfruit suitability was assessed using our fuzzy set model using climate projections of precipitation and temperature for the RCP 4.5 and RCP 8.5 climate scenarios (Fig 4). The results demonstrate successive increase in suitability in the RCP 4.5 and RCP 8.5 scenarios compared to the current suitability (Table 3). At all levels of suitability, our model predicts an increase of total cultivable land (an increase of 27% and 89% for RCP 4.5 and 8.5 respectively) as well as a total increase in average suitability (an increase of 14% and 45% respectively). While a large portion of the total increase in habitat occurred under marginal suitability (<30), increases in the cultivable area of all suitability classes is clear (Table 3).

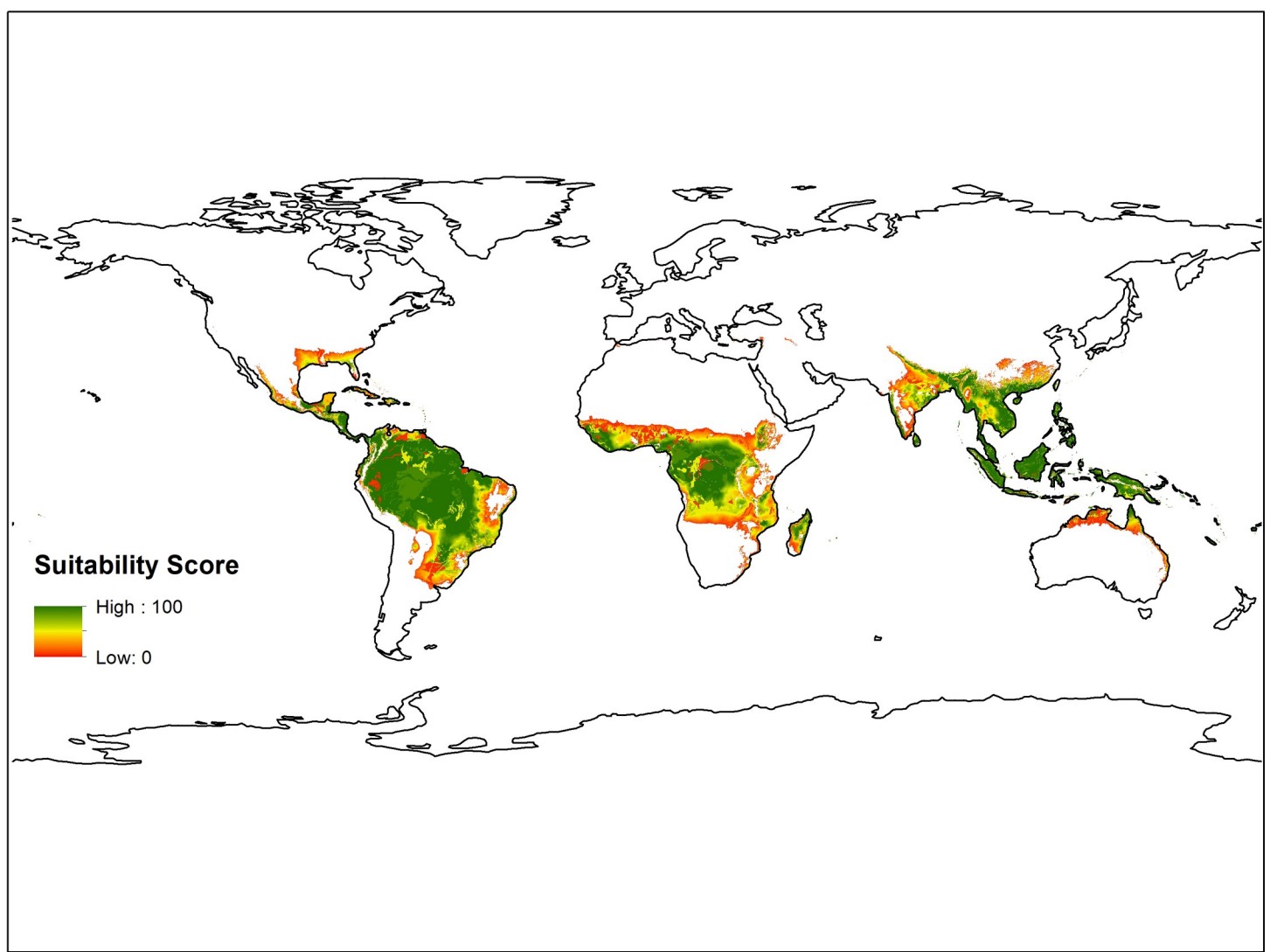

**Fig 2. Current global breadfruit suitability extrapolated from the environmental parameters derived in Hawai'i.**

While our model suggests a net increase in total suitable land and general suitability over the next 50 years, the changes are not uniform and some areas do show a decrease in suitability, including total loss of cultivation (Fig 5). In particular, losses in suitability occur in currently suitable areas of Central and South America, while large gains are seen in southeast Asia, southeastern United States, and southeastern parts of South America.

In general, though, currently productive areas remain largely unchanged. Under the 4.5 scenario, the area that does not change is 43% of the total modelled area (~16,700,000 km$^2$), and the area that increases/decreases by a suitability of up to 10 is a further 35%, making the total area that remains unchanged or minimally unchanged 78% of the total modeled area (~27,400,000 km$^2$). Under the RCP 8.5 climate scenario, the area that will remain unchanged over the next 50 years is 43% of the totaled modelled area (~15,200,000 km$^2$). The area that will minimally change (+/- 10 suitability score), will be another 30%, to make the total area unchanged or minimally unchanged 73% of the total modeled area (~25,500,000 km$^2$).

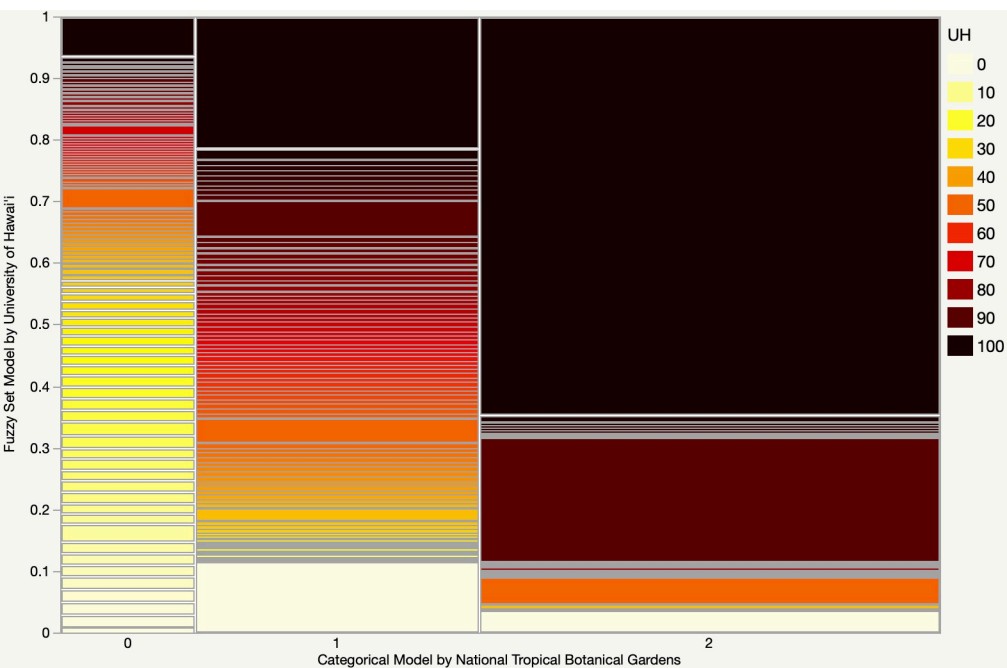

**Fig 3. A mosaic plot comparing a previous global breadfruit model by Lucas and Ragone [62] on the x-axis in categories of unsuitable (0), suitable (1), and best (2), and our fuzzy set model on the y-axis represented by percentage total of each score.** The width of the x-axis indicates the relative total area of that category, while the y-axis indicates the percentage of that area occupied by each suitability score.

## Future opportunities

There is substantial potential for future breadfruit production based on the large increases in total cultivable area and average suitability of that area under climate change projections. This is encouraging given that many current staple crops are expected to decline in suitability with projected future climate scenarios [34, 76–78]. Furthermore, most currently producing regions are not negatively affected, providing some security and stability in the face of projected changes.

The vast growth in very low suitability in our model is a facet of outlier GCM models. All of the very low (<1) suitability numbers result from pixels where only one of the 17 GCMs indicated conditions suitable for breadfruit. Further examination into the inter-GCM model variability would provide stronger confidence in future habitat and risk mitigation investment. This is particularly important for a neglected crop species as there is already substantial risk because of the large scale-lack in agronomic research, post-harvest research, and market development.

As mentioned, our model indicates expansion of area in all bins of suitability. In our observation of validation sites, we noted that suitability of 30 or below represented very poor producing trees–the plants would grow and bear fruit but only on the order of 10% of what trees in very high suitability are able to produce. In this light, we propose excluding the sites with suitability scores less than 30. Similarly, at the upper end of the spectrum, sites with suitability of >70 appear to all be highly productive, with only slight changes in relatively high yields. Therefore, a more tempered approach might be to think of "moderate" (31–70) and "high" (71–100) quality habitat for breadfruit. Such an approach would also inherently eliminate the vast areas of very low suitability (<10) caused by model outliers. Applying this breakdown

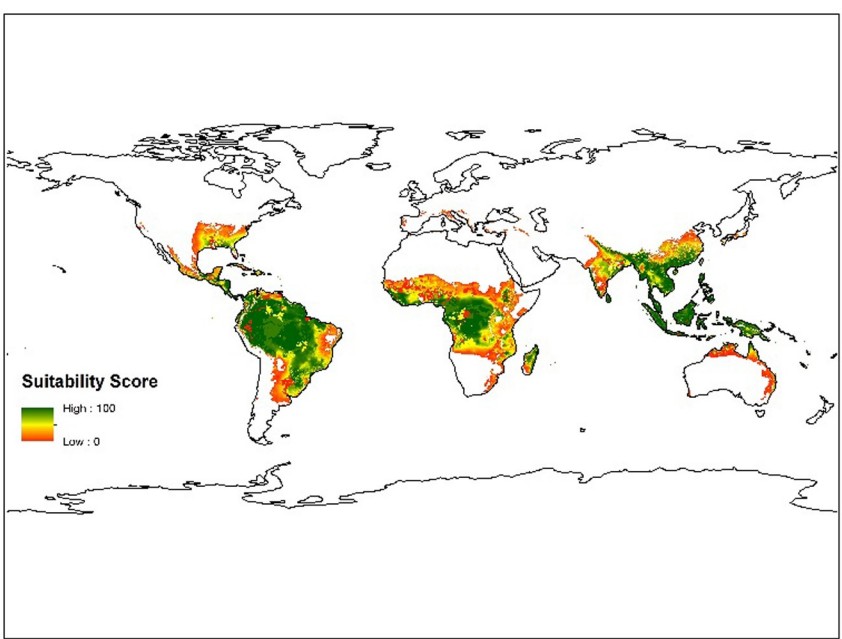

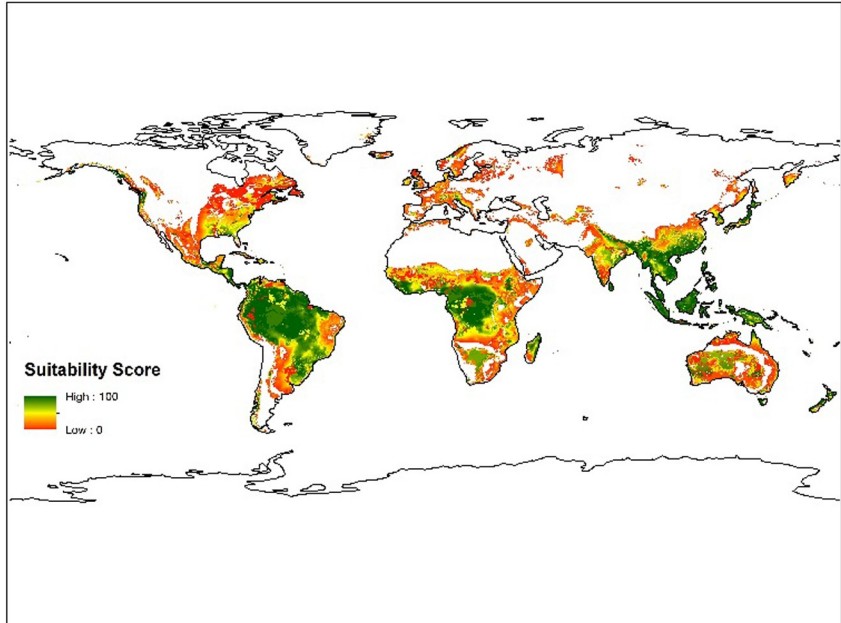

**Fig 4. Modeled future global suitability in 2070 for RCP 4.5 (top) and RCP 8.5 (bottom).**

(Table 3), we see a 10% and 30% increase in high suitability sites for RCP 4.5 and RCP 8.5 respectively, and a corresponding increase of 21% and 90% in moderate suitability.

A current shortcoming to the model is that average annual data was used. This may be particularly problematic in areas that get below freezing temperatures, as frost has been reported to kill breadfruit trees. While the absolute minimum average annual temperature was set at 17°C, freezing temperatures for short periods of time could still occur and be offset by much higher temperatures. Likewise, this could apply to seasonality of rainfall as there is not a clear

**Table 3. Area of breadfruit suitability in millions of km2 for current and future climate projections, presented as total and individual bins as discussed.**

|  | Current | RCP 4.5 | RCP 8.5 |
|---|---|---|---|
| **Total** | **32.66** | **41.62** | **61.71** |
| **Bin** |  |  |  |
| <1 | 2.59 | 3.53 | 9.35 |
| 1–10 | 2.11 | 4.68 | 10.38 |
| 11–20 | 1.97 | 3.62 | 6.20 |
| 21–30 | 1.83 | 2.90 | 5.28 |
| 31–40 | 2.03 | 2.50 | 4.62 |
| 41–50 | 3.27 | 3.74 | 5.17 |
| 51–60 | 1.64 | 1.90 | 3.16 |
| 61–70 | 1.42 | 1.96 | 2.93 |
| 71–80 | 1.63 | 2.12 | 4.17 |
| 81–90 | 4.25 | 5.25 | 6.22 |
| 91–100 | 12.50 | 12.94 | 13.59 |
| **Group** |  |  |  |
| <1 TO 30 | 8.51 | 14.73 | 31.21 |
| 31 TO 70 | 8.36 | 10.11 | 15.88 |
| 71 TO 100 | 18.38 | 20.31 | 23.97 |

understanding of how much prolonged drought breadfruit trees can tolerate. Furthermore, there may be temporal aspects of such variations in weather that further complicate the interactions. For instance, local seasonal drought during the season of vegetative growth may have different impacts than if it occurred during the fruiting season. Unfortunately, there currently exists a drastic shortage of such observations, with the limited observations tending to exist as anecdotal evidence rather than quantifiable parameter thresholds. However, with increased plantings globally and increased focus on the crop, the necessary data to move towards more refined models driven by increasingly specific criteria can be generated. Any future modeling efforts should certainly take an approach that considers monthly extremes in temperature and rainfall as well as rainfall seasonal distributions. However, for the tropical and subtropical regions of the world this shortcoming is not expected to impact the results. Other extreme events, such as heat waves, prolonged droughts, damaging floods and hurricanes, etc. are also not considered in examining suitability in this way.

## Conclusion

Our study highlights breadfruit as a highly resilient tree crop, suitable for investment and incorporation into climate adaptation and regional land-use planning. The dramatic increases in global suitability shown by our model for breadfruit also begs the question of what other NUS crops may flourish under future climate conditions, and can they do this synergistically in an integrated, adaptive food forest. To plan for future adaptation, they must be identified and nurtured now, and supported with technical and infrastructural resources. Approaches such as validated models can be a first step in this direction, providing an increased degree of security and investor confidence to develop the plantings and infrastructure needed. Further modelling that integrates environmental variability will assist in this capacity and where globally available data or high-resolution data is lacking, site specific, fine-scaled models may serve to fill the gap especially for seasonally variable areas.

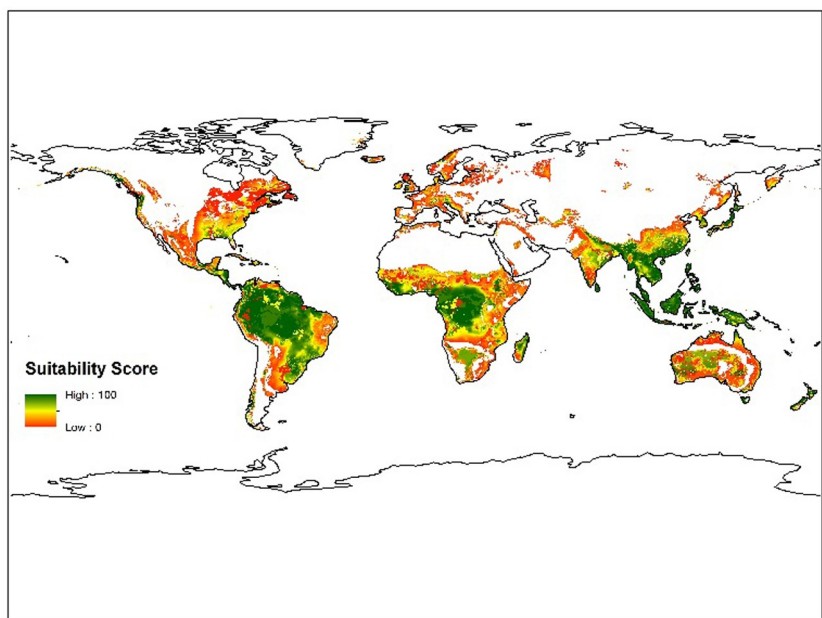

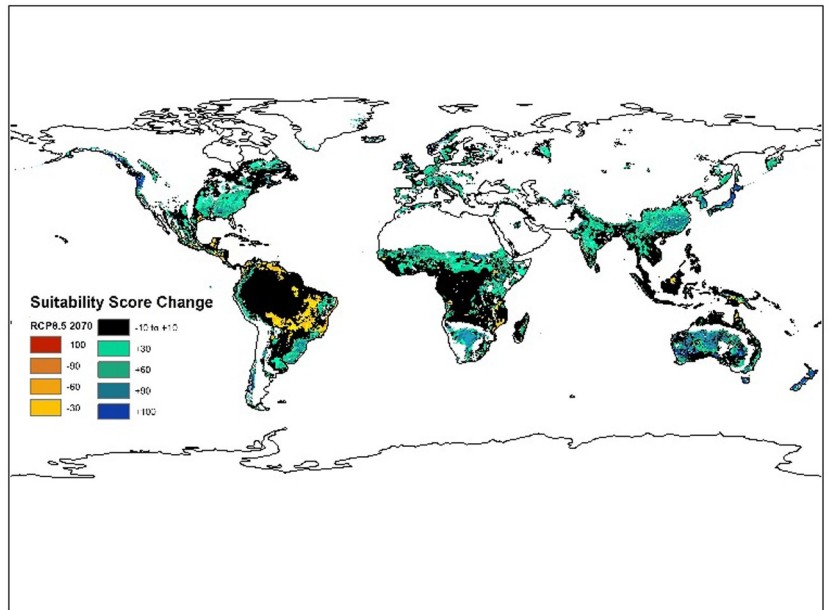

**Fig 5. Changes (increase/decrease) in breadfruit suitability over the next 50 years under (top) RCP 4.5 and (bottom) RCP 8.5.**

Bringing an underutilized crop into the market as a major food source is a difficult task that face many hurdles but is a critical component for addressing mounting needs for food security and good nutrition in a changing world. Our extensive regional work with breadfruit agriculturalist shows a strong need to shift consumer preferences, grow peoples' palette for new crops, and change perception and markets. Large scale marketing efforts may do well supporting this up-and-coming "superfood", especially in light of its ability to persist in the future along with the multitude of economic and environmental co-benefits that may ensue from the farming of sustainable, breadfruit forests.

## Author Contributions

**Conceptualization:** Kalisi Mausio, Tomoaki Miura, Noa K. Lincoln.

**Data curation:** Kalisi Mausio.

**Formal analysis:** Kalisi Mausio, Noa K. Lincoln.

**Funding acquisition:** Noa K. Lincoln.

**Investigation:** Kalisi Mausio, Tomoaki Miura, Noa K. Lincoln.

**Methodology:** Kalisi Mausio, Tomoaki Miura, Noa K. Lincoln.

**Project administration:** Noa K. Lincoln.

**Resources:** Noa K. Lincoln.

**Software:** Kalisi Mausio.

**Supervision:** Noa K. Lincoln.

**Validation:** Noa K. Lincoln.

**Visualization:** Kalisi Mausio, Noa K. Lincoln.

**Writing – original draft:** Kalisi Mausio, Noa K. Lincoln.

**Writing – review & editing:** Kalisi Mausio, Tomoaki Miura, Noa K. Lincoln.

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
