## [Decision Letter · Decision Letter 0]

6 Nov 2019

PONE-D-19-26938

Growing Global Potential of Breadfruit Cultivation under Climate Change Scenarios

PLOS ONE

Dear Dr. Lincoln,

Thank you for submitting your manuscript to PLOS ONE. After careful consideration, we feel that it has merit but does not fully meet PLOS ONE’s publication criteria as it currently stands. Therefore, we invite you to submit a revised version of the manuscript that addresses the points raised during the review process.

We would appreciate receiving your revised manuscript by Dec 21 2019 11:59PM. To enhance the reproducibility of your results, we recommend that if applicable you deposit your laboratory protocols in protocols.io, where a protocol can be assigned its own identifier (DOI) such that it can be cited independently in the future. For instructions see: http://journals.plos.org/plosone/s/submission-guidelines#loc-laboratory-protocols

We look forward to receiving your revised manuscript.

Kind regards,

Ahmet Uludag, Ph.D.

Academic Editor

PLOS ONE

Journal Requirements:

Additional Editor Comments:

Two reviewers ended up with different decisions. I agree with reviewer decided for major revision. I am sure your paper will be better if you follow recommendations but you feel free not to follow not relevant ones but we need your reasoning in that case. I would like to have this paper published. Good luck.

Reviewers' comments:

Reviewer's Responses to Questions

**Comments to the Author**

1. Is the manuscript technically sound, and do the data support the conclusions?

Reviewer #1: No

Reviewer #2: Yes

2. Has the statistical analysis been performed appropriately and rigorously? 

Reviewer #1: No

Reviewer #2: Yes

3. Have the authors made all data underlying the findings in their manuscript fully available?

Reviewer #1: Yes

Reviewer #2: Yes

4. Is the manuscript presented in an intelligible fashion and written in standard English?

Reviewer #1: No

Reviewer #2: Yes

5. Review Comments to the Author

Reviewer #1: The manuscript needs to be significantly re-conceptualised and re-written so that it can be concise, have logical flow and clarity of purpose in terms of contribution. For example, the introduction is too long and winding, and makes for confounding reading. The methodology is then too brief, and the Results and Discussion sections are even more brief with a one sentence conclusion. This is probably because the analyses done is thin and limited; however, a reconceptualisation of the manuscript as a systematic review and metaanlyses, could improve the quality and flow. I have made comments in the manuscript

Reviewer #2: Well written paper with significant research findings for Breadfruit cultivation.

line 57, add a 'comma' between spectrum, obesity

line 184, add the word 'and' between sets, and 5)

line 214 define R studio (add company of the software)

line 225, add the word 'The' between parameters. The model

line 271 add the word'or' between irrigated or amended soils

The 5 environmental criteria selected were rainfall, avg temperature, solar radiation, soil drainage, and soil PH.

The authors discuss the fact that using the avg temperature is one of the shortcomings of their model in that the average temperature does not monitor temperatures below freezing (monthly extremes) which would kill a breadfruit tree. Did the authors also consider that the rainfall (annual precipitation) may also need further refinement? Annual precipitation may occur over 12 months (some rain every month) or it could be characterized as over a short period of time as in areas with monsoons (seasonal rainfall). Would the breadfruit tree be able to withstand a prolonged period of drought? Perhaps the authors would add this discussion in the paragraph from line 389-397.

6. PLOS authors have the option to publish the peer review history of their article (what does this mean?). If published, this will include your full peer review and any attached files.

Reviewer #1: No

Reviewer #2: No

---

## [Author Response · Author response to Decision Letter 0]

27 Dec 2019

Reviewer #1: 

The manuscript needs to be significantly re-conceptualised and re-written so that it can be concise, have logical flow and clarity of purpose in terms of contribution. For example, the introduction is too long and winding, and makes for confounding reading. The methodology is then too brief, and the Results and Discussion sections are even more brief with a one sentence conclusion. This is probably because the analyses done is thin and limited; however, a reconceptualisation of the manuscript as a systematic review and metaanlyses, could improve the quality and flow. I have made comments in the manuscript

We disagree that this paper would be better conceptualized as a review paper as it is based on several years of original, empirical research on breadfruit habitat and productivity, uses such data to calibrate a new habitat model, and then applies that model to existing GCMs to develop a robust picture of the current and future habitat suitability of a long-lived and neglected crop. We would further argue that our research lab is among the few, if any, teams that have the empirical data to calibrate such a model. While we do rely on our previously published data to reduce the need for describing those methods and data, we are not “reviewing” our own work, but building upon it. However, we do agree with the Reviewer that the overall flow and allocation of space in the manuscript was not well developed in the initial submission and have made substantial changes accordingly. The Introduction has been shortened and edited to create better flow and expanded to include previous work of other models. The Methodology has been developed with additional statistical analysis. The Results and Discussion has been combined into one, more comprehensive section, and the Conclusion is much improved. 

Line 44. The introduction is too long and winding and contains a lot of text that does not need to be in an introduction. There may be need to revise the format of the introduction

As stated above the previous Introduction has been edited to be more concise and create better flow, while additional aspects of the introduction have been expanded to include previous work of other models.

Line 46. The extent of literature review on NUS and reports of their potential and limitations is limited. Also, there are some useful papers on climate change impacts on nutrition of fruits and vegetables (dominant ones), that needs to be included.

We have included additional work of models on NUS, although in our search we could not find work examining a long-lived tree crop, which is a major limitation in working with breadfruit and what makes most crop models that are driven by short term weather and physiology not applicable to the modeling of a tree crop. 

A single line to the introduction was added that acknowledges that climate changes will also affect nutritional quality, with two supporting references. 

Line 90. Why is breadfruit still underutilized despite the recognition from ITPGRF and others? What other barriers exist for the crop?

The reasons for underutilization are similar to all NUSs, which were already enumerated in the previous paragraph on NUSs. No additional changes were made. 

Line 98. Can you provide exact quantities to allow for comparative analyses

Providing the exact nutrient quantities for the crop are well outside of the focus of this paper. Multiple references for the statement are provided, including an in-depth review paper on breadfruit nutrition, if readers want to explore the topic in more depth. As we do not explore or discuss breadfruit nutrition in the paper we do not see this as a relevant topic to provide detailed information on. 

Line 115. This could have been strengthened by a more in-depth but brief review of the literature on modelling NUS

We agree, and have heavily edited the first paragraph of this section, and included an entirely new paragraph after conducting some review of this literature. We conclude, however, that existing models driven by daily weather and plant physiology are not applicable in the case of a long-lived tree crop, and think that this supports our habitat modeling approach for modeling breadfruit. 

Line 122. AquaCrop has been applied for several NUS.

Thank you for the suggestion, and we have included some of this literature as mentioned in the previous comment although we conclude that the approach is not currently feasible for breadfruit. 

Line 135. “Than”

Thank you for catching this. Change made. 

Line 153. Is this desirable?

We did not, and are not, making a subjective argument about the desirability of agricultural form here – we are simply reporting details of the study system that we are working within. No change was made.

Line 176. there is no section on statistical analyses. Also, you need to use more than just R^2.

We agree that our statistical analysis of the model performance was inadequate. We have reviewed relevant literature and included three common model performance metrics in order to assess the validation of the model, including the model efficiency, the index of agreement, and the ratio of the RMSE-observations standard deviation. We have included a statistical analysis section in the methods, and include those statistics in the results/discussion section. 

Lines 182-183. I would not consider this stage to be part of mechanistic crop modelling

I think the difference in perspective arises because we are conducted a calibrated crop habitat suitability model that ultimately performs very well as a predictive crop performance model that ultimately performs as a mechanistic crop model (that is, explaining phenomenon in purely deterministic terms). Perhaps the confusion is that our model uses methods that traditionally were applied to habitat suitability and based off of longer-term environmental parameters (climate vs. weather) compared to the plant physiology driven crop models that the Reviewer seems to be more familiar with (Aqua-Crop, etc.). 

Lines 187-188. where these linked to any particular location as these tend to vary by variety and location?

There are some interactive effects of climate that could result in slight differences in the environmental limitations. For instance, the temperature limitation may vary as humidity changes. However, as the EcoCrop data base itself, we apply the same parameters to everywhere in the globe. We are only able to empirically test the performance of breadfruit in a few areas so it is not currently possible to delve into these interactions. We also are not able to pull apart varietal effects at this point, although we have established the long-term trials necessary to do so and hope to refine this model in the long term; however, because of the tree crop timeline we anticipate that this will be on the order of a decade to get the meaningful data needed to appropriate parameterize the model for specific varieties. No changes were made.

Line 200. Can you justify the selection of these criteria and exclusion of common ones such as texture, depth, slope etc

The selection of criteria was based on (1) those available on EcoCrop (2) the panel of experts and distribution of novel trees, and (3) the availability of data. Our intent is not to maximize criteria, but rather to focus on the key criteria that our panel of experts (who are informed by some of the leading agronomic work on breadfruit in the word) suggest. To this end, some criteria are not considered as impactful (soil depth is not limiting since breadfruit has been observed to grow on volcanic rock, and sand substrates) and some others were not available at the spatial coverage required. Short additions were made in the methods to clarify our selection method.

Lines 214-216. Why did you not use MCDA and AHP? Then you could have had more criteria - nine –

We did not want to further increase the criteria at this time because we are simultaneously developing and defining the optimal and absolute levels of the criteria. Our model adheres by the law of minimum indicating at that each criteria is limiting on its own. Multi-criteria decision making wouldn’t apply because we are not looking for the right combination of criteria, but rather, the variable that is most restrictive. For example, for any given site, if rainfall is limiting then to some extent temperature does not matter even if it in the ideal range for breadfruit growth. There is also limited sites that can be used for parameterization and validation. We believe that this is an excellent suggestion for future work and we conduct more modeling approaches to understanding the performance of this crop in different environments. 

Line 252. There are elements of discussion in this section. I would suggest combining the two into one Results and Discussion section; AND Line 361. As elements of the discussion have already been alluded or preempted in the Results section, it may be good to combine the two sections. Also, on its own, the discussion is not supported by the literature, since its already been used in the Results. There is not much context provided in the discussion, and statements of impact are missing

We agree, and as mentioned we have restructured the paper to merge the results and discussion as suggested. 

Lines 311-313. also you had more criteria

As we mention in the methods, we apply the law of minimums, meaning that the most limiting environmental parameter limits the total performance of the crop. Therefore, additional criteria can ONLY restrict the distribution, not increase it. The reviewer may have a fundamental mis-understanding of our modeling approach.

Lines 363-365. PLEASE SUPPORT THIS WITH REFERENCES

References added. Thank you for catching this oversight. 

Line 409. Surely, you cannot have a conclusion that is just one sentence

Conclusion was expanded.

Reviewer #2: 

Well written paper with significant research findings for Breadfruit cultivation.

We thank the reviewer for the positive response to this manuscript. 

line 57, add a 'comma' between spectrum, obesity

Added

line 184, add the word 'and' between sets, and 5)

Added

line 214 define R studio (add company of the software)

Added

line 225, add the word 'The' between parameters. The model

Added

line 271 add the word'or' between irrigated or amended soils

Added

The 5 environmental criteria selected were rainfall, avg temperature, solar radiation, soil drainage, and soil PH. The authors discuss the fact that using the avg temperature is one of the shortcomings of their model in that the average temperature does not monitor temperatures below freezing (monthly extremes) which would kill a breadfruit tree. Did the authors also consider that the rainfall (annual precipitation) may also need further refinement? Annual precipitation may occur over 12 months (some rain every month) or it could be characterized as over a short period of time as in areas with monsoons (seasonal rainfall). Would the breadfruit tree be able to withstand a prolonged period of drought? Perhaps the authors would add this discussion in the paragraph from line 389-397.

This is an excellent point, and it was considered although not explicitly discussed. We have added a portion to the discussion where this is considered.

---

## [Decision Letter · Decision Letter 1]

21 Jan 2020

Cultivation Potential Projections of Breadfruit (Artocarpus altilis) Under Climate Change Scenarios Using an Empirically Validated Suitability Model Calibrated in Hawai’i

PONE-D-19-26938R1

Dear Dr. Lincoln,

We are pleased to inform you that your manuscript has been judged scientifically suitable for publication and will be formally accepted for publication once it complies with all outstanding technical requirements.

With kind regards,

Ahmet Uludag, Ph.D.

Academic Editor

PLOS ONE

Additional Editor Comments (optional):

Please follow the reviewers small comments. Before your manuscript came to yo for checking, please do all recommended changes. I will acept this paper to accelarate the publishing instead of minor revision.

Reviewers' comments:

Reviewer's Responses to Questions

**Comments to the Author**

1. If the authors have adequately addressed your comments raised in a previous round of review and you feel that this manuscript is now acceptable for publication, you may indicate that here to bypass the “Comments to the Author” section, enter your conflict of interest statement in the “Confidential to Editor” section, and submit your "Accept" recommendation.

Reviewer #1: All comments have been addressed

2. Is the manuscript technically sound, and do the data support the conclusions?

Reviewer #1: Partly

3. Has the statistical analysis been performed appropriately and rigorously? 

Reviewer #1: Yes

4. Have the authors made all data underlying the findings in their manuscript fully available?

Reviewer #1: Yes

5. Is the manuscript presented in an intelligible fashion and written in standard English?

Reviewer #1: Yes

6. Review Comments to the Author

Reviewer #1: I enjoyed reading your responses to the initial comments. I suggest that you remove the sub-headings in the introduction, as they do not seem to add much value. The sub-section that refers to why Hawaii as a location, should be part of the methodology, and not the introduction. The introduction itself, would still benefit from some shortening for improved readability and conciseness. As it is, it is overly long. The discussion could also still benefit from being strengthened - for example, some questions posed in the conclusion could have been answered/addressed in the discussion, as there is published information available to provide such discussion.

7. PLOS authors have the option to publish the peer review history of their article (what does this mean?). If published, this will include your full peer review and any attached files.

Reviewer #1: Yes: Tafadzwanashe Mabhaudhi

---

## [Editor Report · Acceptance letter]

7 May 2020

PONE-D-19-26938R1 

Cultivation Potential Projections of Breadfruit (*Artocarpus altilis*) Under Climate Change Scenarios Using an Empirically Validated Suitability Model Calibrated in Hawai’i 

Dear Dr. Lincoln:

I am pleased to inform you that your manuscript has been deemed suitable for publication in PLOS ONE. Congratulations! Your manuscript is now with our production department. 

With kind regards,

on behalf of

Dr. Ahmet Uludag 

Academic Editor

PLOS ONE